# Promoting Sexuality Education for Children and Adolescents on a Large Scale: Program Design, Organizational Cooperation Network and Administrative Mobilization

**DOI:** 10.3390/children9101474

**Published:** 2022-09-27

**Authors:** Huina Dong, Hongyan Li

**Affiliations:** 1School of Social and Behavioral Sciences, Nanjing University, Nanjing 210023, China; 2Independent Researcher, Beijing 100050, China

**Keywords:** sexuality education for children and adolescents, social organizations, standardized program design, organizational cooperation network, administrative mobilization

## Abstract

In China, the promotion of sexuality education for children and adolescents is hindered by a relatively conservative culture and insufficient drive from the government. With the government and the market failing to deliver in this area, social organizations, such as the third sector, are playing an important role. This paper mainly discusses how Chinese social organizations promote sexuality education for children and adolescents on a large scale. This paper studied NW, the largest sexuality education support platform in China at present, and 16 frontline implementing organizations in its cooperation network. This research collects empirical data through participatory observation and semi-structured interviews, involving a total of 37 interviewees, plus relevant text data. The study follows the analytical method of process tracking, trying to extract the key mechanisms of how social organizations promote sexuality education from the processes demonstrated by the specific cases. This paper focuses on the key phases of a standardized sexuality education program, from its design to promotion and then to implementation. It is found that, based on the large-scale operation strategy, the operation process of the sexuality education program exhibited three core mechanisms: standardized program design, organizational cooperation network and administrative mobilization. These three mechanisms have become the key force to break through many of the cultural and institutional obstacles and achieve large-scale implementation of sexuality education. Finally, this paper discusses the challenges of operationalizing the sexuality education program, with compromised teaching quality as a major one, and points to the direction for future research.

## 1. Introduction

In recent years, sexuality education for children and adolescents has become an important social issue, drawing extensive attention from the Chinese society. However, the popularization of sexuality education in China still faces many obstacles. First, from the social perspective, the related attempts to promote sexuality education have encountered cultural taboos and public opinion disputes [1,2]. In addition, the concept of sexuality education originated from the West and is incongruent with traditional Chinese culture and practices in terms of sex [3,4]. Secondly, the development of sexuality education courses and educational products also requires a considerable amount of financial and time investment, and its cost-effectiveness is difficult to assess, which stymies the involvement of the private sector. Thirdly, although the issue of sexuality education for children and adolescents has been mentioned many times in the national policies, the reality is that local governments pay little attention to it. The main reasons why the local governments ignore the requirements of macro-policies include the weak incentives and supervision provided by the central or higher-level government in terms of sexuality education, and the lack of clear performance indicators and relatively high policy implementation risk in this area [5]. In addition, before the newly revised Law of the People’s Republic of China on the Protection of Minors came into effect in 2021, the term “sexuality education” was avoided in more than 30 important policy documents concerning children and adolescents, most of which used terms such as puberty education and health education instead [6]. While it is difficult for the market and government to effectively provide sexuality education services, social organizations, which are regarded as the third sector, are playing an important role in promoting sexuality education, making up for government and market failures by providing public services and solving social problems [7].

The huge gap between the demand and supply of sexuality education makes it difficult for a single social organization to effectively address the problem. Therefore, social organizations tend to adopt a large-scale operation strategy. Around 2015, foreign concepts such as “scaling social impact”, “social innovation” and “social enterprise” were introduced into China. These concepts tried to introduce the market-oriented model into the public welfare field and promote large-scale solutions to social problems [8,9]. Under this influence, the concept of scaling up was explored in China’s public welfare field [10,11,12]. The large-scale development strategy in the public welfare field requires social organizations to formulate standardized and replicable public welfare products/programs, addressing the social issues of their concern. Once the products/programs are proved effective in implementation, they are delivered to more regions to serve more people and solve social problems on a larger scale [13]. It should be noted that the focus of scaling up is not the significant expansion of a single social organization, but the scale in terms of the promotion of its public welfare products/programs and the impact generated.

The fact that the large-scale operation strategy has gradually become the mainstream concept of the public welfare sector in recent years can be largely attributed to the gradual formation of the public welfare ecosystem. Researchers have constructed the concept of the “public welfare ecosystem” which includes beneficiaries, public welfare organizations, capacity builders, regulators and communities [14]. However, this concept only describes the division of labor among different public welfare entities, and does not deeply discuss its structural and functional characteristics [15]. In the recent research on China’s public welfare ecosystem, some of the scholars have discussed the functions of different subjects in the ecosystem. It is believed that social organizations’ participation in solving the social problems depends on the support of foundations and supportive platform organizations that provide the funds and program design in a top-down approach. The participation of different social actors can help enhance the impact and resource mobilization ability of the social organizations, promote policy changes and solve social problems more effectively [16,17].

At present, China’s social organizations mainly focus on the agendas that are in line with the government’s priorities, such as care for the aged, poverty alleviation, disaster relief and children’s education [18]. Sexuality education remains a relatively sensitive and marginal social topic, with considerable challenges in its promotion. This paper adopts the qualitative research method to better understand the promotion of sexuality education by social organizations, and to explain how this marginal topic can be promoted through a top-down operation process among social organizations. In this paper, the operation process of the sexuality education program is divided into three key phases: design; promotion and implementation. For each phase, some key questions are put forward: How do social organizations understand children’s and adolescents’ sexuality education and develop corresponding sexuality education programs? How do different types of social organizations establish cooperation to jointly promote sexuality education? How do frontline social organizations integrate the agenda of sexuality education into the local society? Finally, this paper discusses the actual operation effect of the sexuality education program, and summarizes the experiences and limitations of social organizations in program implementation, which is of great significance to the continuous and in-depth promotion of sexuality education in China.

## 2. Materials and Methods

### 2.1. Research Methods

This paper mainly adopts the method of process tracking. Based on the inference within a case, process tracking is a tool for investigating the causal mechanism in case studies [19]. Social organizations’ promotion of sexuality education is a process based on their located social culture and institutions, and dominated by the internal key factors which continue to function. Therefore, differing from the analysis path of general case studies, which first forms an analysis framework and is then confirmed by specific cases, this paper follows the research paradigm of process tracking, to extract the key mechanisms demonstrated by the process of sexuality education promotion by social organizations, to explain how social organizations can promote sexuality education on a large scale.

### 2.2. Research Subject

This research focuses on NW for the case study. NW is a non-profit organization dedicated to providing high-quality sexuality education services for children and adolescents aged 6–18. The sexuality education program of NW was initiated in 2016, with a gradually formed goal of implementation at scale. NW tries to reduce the barriers and costs of implementing sexuality education by constantly innovating its service delivery model, so as to allow more children and adolescents to access quality sexuality education. At present, NW has become one of the most professional and influential sexuality education providers in China, and it is also one of the forerunners among social organizations that explicitly put forward a strategy for large-scale development.

### 2.3. Data Collection

This paper mainly uses semi-structured interviews, participant observation and text data collection to obtain the empirical data. The first round of research was conducted from 1 September 2020 to 30 October 2020 by the first author who participated in the daily work of NW as an intern so as to obtain a deeper understanding of the scaling up of its sexuality education program. During the internship, the researcher obtained a large number of written materials, including meeting minutes, statements of financial accounts and various technical materials of sexuality education developed by NW. In the first phase of the research, the researcher conducted offline and online interviews with a total of 16 people from NW and its partner organizations engaged in the sexuality education program: six employees of NW; four project leaders of provincial hub organizations; five leaders of local social organizations and one employee who worked for a well-known private foundation in China responsible for evaluating large-scale programs.

The second round of research was conducted by the two authors from 18 April 2022 to 20 May 2022, when the researchers interviewed the heads of 11 local partner organizations in the cooperation network of NW. Altogether, the two rounds of research interviewed representatives of 17 social organizations including NW (see Table 1), and four of these organizations were interviewed twice. In addition, through the facilitation of the local social organizations, the researchers also conducted interviews with three government officials and three sexuality educators trained as sexuality educators from local school teachers, which informed the understanding of the perspectives of local governments and schools.

Besides the two rounds of interviews, the first author also visited local social organizations from 1–9 July 2021, participated in the training activities organized by them and observed their teaching in practice. So far, the authors have investigated 17 social organizations, involving a total of 37 interviewees. The interviews were conducted in Chinese, each one lasting one to two hours, and were recorded and transcribed. The informed consent forms were signed with each social organization, which authorized the authors to conduct academic research and publish papers using the data. According to academic common practice, the paper strictly follows the rule of anonymity to ensure the confidentiality of organizational and personal information.

## 3. Results

Using the analysis method of process tracking, this paper focuses on the three key phases of the sexuality education program of NW. First of all, NW designs a standardized sexuality education course based on the concept of comprehensive sexuality education, and tries to promote the course implementation by cooperating with other social organizations. Secondly, from the theoretical perspective of the public welfare ecosystem, social organizations at different levels have different resource advantages and play different roles. The social organizations complement each other with their unique resources by constructing a cooperative network, which provides the organizational network foundation for sexuality education implementation across the country. Thirdly, the local social organizations often choose to cooperate with the government in the process, so as to achieve large-scale implementation of the sexuality education program.

The researchers observe that three key mechanisms have emerged In the process of scaling up sexuality education: standardized program design; organizational cooperation network and administrative mobilization. These three mechanisms have become the key to breaking through many of the cultural and institutional obstacles and realizing the large-scale promotion of sexuality education.

### 3.1. Standardized Sexuality Education Program Design

NW attaches importance to the research and development of the sexuality education products, believing that a low-threshold, standardized and replicable sexuality education course is the foundation for realizing the large-scale promotion of sexuality education. In China, sex is generally regarded as a scourge, something that is negative and unspeakable. There are also abundant stereotypes about gender roles, which are contrary to the concept of comprehensive sexuality education (CSE), conceptualized by The United Nations Educational, Scientific and Cultural Organization (UNESCO). NW designed its sexuality education course in line with the concepts and content framework of CSE [20], as proposed by the International Technical Guidance on Sexuality Education published by UNESCO in 2018.

CSE involves the teaching and learning of the cognitive, emotional, physical and social aspects of sexuality. It aims to provide comprehensive, scientific and fact-based accurate information to children and adolescents in culturally relevant and age-appropriate ways, while cultivating the necessary skills and attitudes/values [21]. CSE has been accepted and implemented in many of the countries across the world. Previous research shows that CSE could not only improve the development of children’s sexuality, such as the acquisition of sexual health knowledge, but also positively impact children’s interpersonal attitudes and contribute to their social development [22,23].

In line with the concept of CSE put forward by UNESCO and referring to its content framework, NW fully adopts the age-appropriate principle, and takes into consideration the current school curriculum arrangement when designing the course; thus, developing a standardized sexuality education teaching package for primary and secondary schools. The course was divided into four age-appropriate segments: lower primary; higher primary; junior high school and senior high school. Following an iterative process of development, the course now includes 13 lessons for primary school and 15 lessons for middle school. Each lesson lasts 40 to 45 min and consists of one or two short animated videos and two to three participatory activities. The accompanying teaching package consists of three basic standardized documents: PowerPoint slides to be used in class; lesson plans to help the teachers with teaching preparation (providing teaching methods and details of procedures) and recorded videos of a teaching demonstration of each lesson. Using animated videos to convey the core information not only facilitates students’ interest in learning, but also ensures that the content of sexuality education received by students is consistently scientific and accurate, thus minimizing the problem of compromised teaching quality due to the capacity gap among the sex educators. As previous studies have revealed, sex educators face some obstacles in delivering sexuality education, such as the lack of adequate teacher training and of high-quality courses and materials [24,25]. In addition, according to the interviews in this research, the sex educators may also face the pressure of public opinion and opposition from schools and students’ parents.

Teacher Hu, who was a sexuality educator trained by the organization O and at the same time a primary psychology teacher, told us, “Nowadays, parents and schools are not very receptive to sexuality education. They are scared. On social media, I noticed some parents were opposed to sexuality education. I was worried about how my teaching career might be affected if I start teaching sexuality education…” “But some teachers asked me privately if I could teach the fourth and fifth graders about puberty…” Teacher Hu later found that her sexuality education class was very much welcomed by the students, and the school and parents were supportive of this. She said, “Sexuality education is a novel subject and the children find it attractive. They listened very carefully in class and were attentive in watching the animations”. Her students were very excited when seeing her at the monthly sexuality education class, hugging her and saying that “I missed you so much”. Teacher Yang also told us that some students would voluntarily approach her after class with questions about menstruation. These examples show that the delivery of sexuality education enables students to feel that they are being cared for and valued, thus improving the teacher–student relationship.

She told us, “Once a parent told me that, her child now takes a bath every day because I taught the students that they should clean their genitals every day. She said even on cold winter days, the child would still insist on doing this and would not compromise on doing it every other day. It surprised me to see how the students really learned and also applied the learning in their daily life”. We came across many similar examples during the interviews. It is worth noting that, unlike adults, children are not averse to sexuality education but instead show a lot of curiosity, desire and delight in learning. The timely receiving of CSE has generated a positive impact on the growth of children, as well as the teacher–student relationship.

It is obvious that the large-scale promotion of sexuality education is inseparable from the availability of trained educators; it is necessary to motivate the educators to face difficulties bravely and dare to challenge the conservative gendered culture and ideas [26]. Therefore, besides developing a standardized sexuality education course, NW also attaches equal importance to training sexuality educators through a standardized teacher training program, which they hoped could help to minimize the psychological burden as well as the time and financial investment of the educators, to develop a group of qualified volunteer sexuality educators in a relatively short period of time. This training played an important role in changing the attitudes of the sexuality educators. The responsible persons of the social organizations we interviewed shared with us the changes they experienced after attending the training. The head of the organization H is a young man who had never been involved in sexuality education before. He told us, “In fact, I had very vague ideas about sexuality education in the beginning. I knew nothing about it. I was embarrassed when I first joined the training. But later I discovered a new world, and my views experienced a fundamental change”. “We tended to feel shy and embarrassed when talking about sexuality. We thought it was wrong to talk about it, but in fact it is normal. If we adults are ignorant about our bodies, how could we expect to inform our children? It is very sad”. The person in charge of the organization J also changed her stereotypical and biased understanding of sexuality education after receiving the training. She said, “We used to do sexuality education with the main focus on preventing sexual abuse, but after the training, I found this inappropriate, as it might give the impression that sexuality is only related to something negative”.

Obviously, the awareness and teaching abilities of the school sexuality educators directly affect the actual result of sexuality education. On one hand, when the school leaders and teachers fully recognize the significance of sexuality education, the schools can make a difference in the allocation of teachers and class hours. One teacher even said that she had delivered all of the lessons in the course package and hoped that NW could develop more lessons for teachers to choose from. She also expressed a strong personal desire for continuous learning. Some teachers go beyond the limitations of the classroom and campus and see a wider range of social structural problems affecting students; their educational ideas have also changed significantly. For example, the head of organization L, who is also a primary school teacher, undertook social work outside the school to help children in distress as part of her post-graduate research project, which has brought her a new perspective on student engagement. With her expanded experience in her dual roles, she was able to timely detect the needs of students and help them solve related problems by linking them with the various resources offered by the government and the community.

On the other hand, transformation is never easy. Through the interviews with the sexuality educators in schools, it was found that the teachers preferred a three-day sexuality educator training program, as it helped them understand CSE and gain relevant teaching skills and methodologies in a short time. However, it appeared that the one-off training was not enough for the teachers to fully understand the concept and connotations of CSE and embody it in their teaching practice. In the interviews, the authors noticed that some of the school sexuality educators habitually resorted to the conventional cultural biases. For example, according to a school teacher, when communicating with a junior high school girl about puppy love, she emphasized the negative effects of having physical contact with boys, such as the impact on her studies, damage to her reputation and disapproval from parents. Apparently, the teacher was not aware that this kind of scare and shame strategy is not conducive to guiding students into developing appropriate sexuality and gender-related values and behaviors.

Instead of seeking to design a top-notch sexuality education course, NW gives priority to making the course user-friendly for wider adoption. NW believes that only the standardized products that meet students’ educational needs have a basis for replication and expansion across the country. As the head of NW said, “A highly skilled expert can never win over a system. Some organizations have developed products that they claim to be sustainable and replicable, but they have not been truly recognized and widely adopted. For example, the sexuality education courses developed by some of the leading scholars are excellent with very comprehensive content, but because of the complicated design and the demand for dozens of class hours to deliver, it is hard to be implemented in schools”. Therefore, NW hopes that a highly standardized, replicable and practical sexuality education program can allow ordinary teachers, social workers and even volunteers in social organizations to deliver sexuality education with relative ease, which will help promote sexuality education in schools regularly and systematically.

### 3.2. Construction and Operation of the Social Organization Cooperation Network

#### 3.2.1. Value Orientation and Complementary Resources

The large-scale promotion of the sexuality education program can be enabled by social organizations as the main actors, through a program chain of “sexuality education program-social organizations-educators-learners”. The choice of NW to seek cooperation with local social organizations to jointly promote sexuality education is influenced by the institutional environment faced by social organizations and the developmental characteristics of the public welfare sector itself.

The formation of this organizational cooperation network is influenced by China’s current institutional environment. Article 17 of China’s Regulations on the Registration and Administration of Social Organizations stipulates that social organizations are not allowed to set up regional branches. Therefore, although NW is committed to the cross-regional promotion of sexuality education, it cannot promote its program by establishing regional branches, that is, expanding the organization itself. Moreover, the establishment of branches means huge management and manpower costs, which are not affordable for social organizations such as NW with weak financial capacity. Therefore, NW chose to promote sexuality education by building a cooperative network with other social organizations.

At present, according to the division of labor among Chinese social organizations, local social organizations are mainly implementing specific projects with support from foundations and other supportive social organizations that design the public welfare projects and provide overall management support. The local social organizations include both hub organizations and frontline implementing organizations. The hub organizations generally act as intermediaries between the foundations, supportive social organizations and frontline implementing organizations. With relatively stronger resource mobilization ability and local influence, the hub organizations receive resources from upstream and pass them downstream to the frontline implementing organizations. Initially, NW chose to build a cooperation network of frontline implementing organizations through provincial hub organizations, but later, in order to further scale up the sexuality education program, they adjusted the cooperation model and started to directly engage frontline implementing organizations to scale up the program more effectively.

This cooperative network enables the different types of social organizations to complement each other’s resources, which facilitates resource mobilization and program scaling up. From the perspective of NW, firstly, the knowledge and technical support it provided to local social organizations met their local needs. The local social organizations usually undertake projects through government procurement schemes to provide all kinds of educational services for children and adolescents, including child poverty relief, sponsorship and mental health services to students in distress. In the process of identifying less visible local needs, the local social organizations noticed many of the negative consequences caused by the lack of sexuality education, especially in the rural areas with a large number of left-behind children as a result of the heavy labor outflow. In addition, there are acute psychological problems among children caused by sexual abuse and a lack of necessary self-protection and health knowledge and skills. These social problems gave the social organizations a sense of urgency to provide sexuality education. Some of the organizations had already tried to develop their own related projects before adopting the NW sexuality education program, but they were faced with various obstacles, such as insufficient professional capacity and a conservative local culture. Therefore, they showed great trust, appreciation and enthusiasm for participating in the NW sexuality education program, which they deemed to be professional. As the person-in-charge of the organization N said, “Even if I spend a year or two, I could not possibly develop a sexuality education course like this. I lack the confidence to propose to the school principal that I want to deliver a sexuality education course in the school. This high-quality course created by a professional team gave me the confidence to provide the teaching through our trained sexuality educators”.

Secondly, apart from the knowledge and technical support, NW also gives support in terms of funding, project management advice and legitimacy endorsement, which helps to dispel the concerns of the local social organizations and inspire their enthusiasm for participation. According to the statistics of the Chinese Social Organizations Public Service Platform [27], the total number of registered social organizations has exceeded 900,000, of which 41.34% are at the county/district level and 29.38% at the municipal level. However, nearly half of these organizations were established for less than five years. With a short history and a lack of experience, they largely depend on the support of government resources. Most of the social organizations in China are in the primary stage of development, struggling to survive with limited resources, resulting in the resource-oriented development approach [28].

With the understanding of the approach of local social organizations, NW highlights the following outstanding advantages of its sexuality education program: first, it provides essential financial support to resource-oriented local social organizations; secondly, the sexuality education course is professional, standardized and with a low-threshold, accompanied by a standardized sexuality educator training workshop, as well as project management guidance offered to the local partners, further lowering the participation threshold; thirdly, the local social organizations can make use of their advantages in implementing the sexuality education program, with local schools and communities as their main channels to provide social services for children and adolescents and fourthly, the cooperative relationship between NW and the provincial hub organizations can better meet the multiple needs of local social organizations as the frontline implementers, by providing funds, legitimacy endorsement, branding support, etc. By doing so, a comprehensive system, including technology, branding and resources, was made available and provided much-needed support to frontline implementing organizations for promoting sexuality education.

For example, the provincial hub organization A enjoys a close relationship with the provincial Communist Youth League Committee, which enabled them to successfully push for the release of an official document about sexuality education by the Committee to local government departments. This document provided a political endorsement for the social organizations to conduct sexuality education in the province. As explained by the head of organization A, “The Communist Youth League Committee has a very strong influential power in the government system, and it seldom releases such documents, so the issuing of the document on sexuality education dispelled the concern of local education bureaus and women’s federations, and opened the doors for local social organizations to seek support from them”.

Whether it is the provincial hub organizations or the frontline implementing organizations, they all have local advantages that NW does not have. The person in charge of NW said, “The partnership is powerful. We as a single organization couldn’t possibly cooperate with so many local governments”. Through interviewing the leaders of the other 16 social organizations within the cooperative network, the authors felt deeply about the importance of uniting the local social organizations in scaling up sexuality education. These local social organizations share several advantages: Their funds mainly come from the local government’s procurement; they have close relationships with local government departments and they enjoy certain local influence and popularity. The local governments in China tend to procure social services from locally registered social organizations with experience in cooperating with the government, while providing them with administrative legitimacy, funds and landing channels to support organizational development [29]. In addition, the local social organizations know the local situation better than NW, who, as an outsider detached from the local project operation, neither understands the actual needs nor finds it easy to earn the trust of the local governments. With their unique advantage in the local area, including better local knowledge and more likelihood of winning the trust of local society and governments, the local social organizations are in an advantageous position to mobilize the local government resources and widen their channels for the landing of the sexuality education program.

Therefore, one of the characteristics of the current development of China’s public welfare sector is the increasing differentiation and stratification of social organizations, while the cooperation of social organizations of different types facilitates the resource complementarity to enlarge the impact of the public welfare programs. As a developer and promoter of the sexuality education program, NW highlights its value of providing professional knowledge and sexuality education products, as well as the essential funding support and management capacity building to its local implementing partner organizations. As the head of NW said, “Once the scale of the program reaches certain level, we are more concerned about how to provide more specialized service and maximize our technical resources, trying to achieve greater impact across the sector by focusing on a specific area”.

#### 3.2.2. Construction of a Cooperation Network and Its Effectiveness

Based on the above discussion, NW has defined the path of promoting sexuality education via the organizational cooperation network. Specifically, in the two years from 2019 to 2020, NW carried out in-depth cooperation with four provincial hub organizations in Shandong, Shaanxi, Hunan and Anhui provinces, providing comprehensive support in technology, funds, management and branding. NW proposed the target of 30,000 student attendances [30] per year for each of the hub organizations, with some clear requirements on the delivery quality; for example, the large-class teaching and the direct delivery of teaching by the hub organizations themselves should not exceed 30% of all of the delivered lessons. In terms of funds, NW provided about $9000 to be used on the project implementation and $1500 as a study grant for each hub organization every year. As to branding, it encouraged hub organizations to create their own branding for the sexuality education and helped them with branding promotion to raise their popularity and influence in the local areas. With the hub organizations as the intermediary, NW does not need to interact with the frontline implementing organizations directly, which saves considerable labor and management costs. Moreover, these four hub organizations have previously cooperated with NW on other projects, so they are familiar with, and have a high degree of appreciation for, the sexuality education program, and are willing to continuously invest resources to promote the program in their respective provinces.

However, at the end of 2020, NW adjusted its cooperation model and directly established closer and more effective cooperation with the local social organizations. Any local organizations with the capacity to achieve 3000 student attendances or more in classroom teaching on sexuality education, either as one single organization or as a network of organizations, can directly partner with NW. At the same time, NW distinguished two kinds of organizations for direct cooperation: basic partners and key partners, who received different levels of support in terms of funds, materials, branding and organizational management. NW further increased the overall capital investment, from funding only four provincial hub organizations to providing direct financial support to the frontline implementing organizations. The enhanced level of financial support is very attractive to the local social organizations that adopt the resource-oriented development approach, and to some extent inspired their enthusiasm for participation. With this adjustment of the cooperation model, NW is no longer preoccupied with managing and cultivating relationships with hub organizations only, but instead has turned to frontline implementing organizations and engaged them as the direct channels for the promotion of sexuality education. This helped to speed up the pace of expanding the program coverage throughout the country.

By the end of 2021, the cooperation partners of NW had spread across 26 provinces. The initial in-depth cooperation with four hub organizations has expanded to more than 200 directly engaged local social organizations. By May 2022, NW had mobilized more than 200 frontline implementing organizations to teach sexuality education in local schools and communities, reaching 3037 schools and involving more than 13,900 volunteer educators and school teachers, achieving over 2 million student attendances in sexuality education classes [31].

### 3.3. Implementation of the Sexuality Education Program

#### 3.3.1. Mobilizing the Local Governments

The adoption of the sexuality education program means that the school accepts the course and the students have the opportunity to attend the classes. As such, the school is a critically important partner in the promotion of sexuality education. In recent years, much more attention has been paid to the impact of the lack of sexuality education on the health and well-being of children and adolescents, and national policies have been improved to address the problem. However, under the education system with test scores as the primary marker of success, sexuality education remains nearly invisible on the school agenda. Because sexuality education remains a socially sensitive issue, schools have concerns that hinder the on-campus sexuality education.

Based on the interviews, the major concerns of schools and teachers include: First, worries that the parents will oppose sexuality education and report to the local education authority that the school teaches “bad things” to their children; Secondly, the school is worried that the sexuality educators from the social organizations are not professional enough to handle the sensitive sexuality education content, and may even mislead children leading to appropriate behaviors. Based on this, many of the schools do not allow the social organizations to teach in the schools. In some cases, the head of a social organization can convince an open-minded school principal to accept the offer. However, in the face of more conservative school principals, the permission is hard to obtain, regardless of the personal relationships. Quite a number of the people in charge of the social organizations mentioned that the schools want to see if there is government support behind the program, and if there is not, even with good personal relationships, the principal will not allow the social organizations to teach in the school. Even if permission is obtained for entry into the school, it is difficult for the social organizations to teach the whole course systematically, because of the limited teaching hours provided by the school. Therefore, relying on personal relationships to promote sexuality education is unsustainable, and will not achieve what NW expects of the program.

Faced with this challenge in entering schools, the local social organizations turn to efforts to mobilize the government. Through this process, the persons in charge of the social organizations have gradually realized the value of leveraging local administrative power to mobilize the schools’ participation in training and onboarding sexuality education. The government can mobilize all of the schools to participate on a large scale, which is far beyond what the social organizations can achieve through personal relationships. This is more likely to achieve the systematic delivery of sexuality education in schools through mobilizing school teachers, who, with greater experience than volunteer educators, are more acceptable for schools. Finally, it is more efficient for the government to organize school leaders and teachers to participate in the teacher training activities, considerably reducing the program’s operation cost to the social organizations. The studies show that, in the authoritative governance system of China, the functions of the social organizations are closely bound by the control and guidance of the state, and the governance will and operational logic of the state and government profoundly affect the operation of social organizations [32].

#### 3.3.2. The Key to the Success of Administrative Mobilization

The 16 local social organizations investigated through this research have achieved varying degrees of success in mobilizing government support for organizing local schools’ participation in sexuality education. For example, the head of the organization J is a local judge. This identity helped her obtain financial support from the local Communist Youth League Committee to carry out sexuality education in many of the schools in the whole district. Another example is the county-level organization F. Since 2019, when the organization started to implement the sexuality education program, it has successfully embedded the project in the work of the local Communist Youth League Committee, by rebranding the project as a civilized social practice in the new era, in line with the government initiative. The head of the organization together with NW was even able to convince the Communist Youth League Committee in the county to accept sexuality education as a pilot practice and demonstration model and brought it to the attention of the upper-level government. As a result, organization F organized a municipality-wide sexuality education training workshop for the school. This success with a bottom-up approach in promoting sexuality education eventually led to nearly 30,000 student attendances in sexuality education classes.

By investigating the reasons behind the successful administrative mobilization by these organizations, this research identifies two key mechanisms that played a role: first, the personal contacts and network resources accumulated by the social organizations and secondly, the project implementation approach and embedding strategies adopted by the social organizations. As far as the first mechanism is concerned, the authors found that the deep cooperation between the local social organizations and the local government basically depends on the personal contacts and network resources of the heads of the local organizations. Those heads who happen to come from relevant government departments have a greater advantage. The head of the organization J is a local judge. The head of the hub organization B is a delegate of the Provincial People’s Congress, and the head of the organization I used to be the party secretary of the local sub-district office. The head of organization F has been working in the county government propaganda office for many years and is very familiar with the government needs and approaches, and hence is good at identifying cooperation opportunities. He said, “I have been working more than a decade in the government, and I know how the (administrative) power operates and the constraints and distribution of rights and interests among different departments of the government”. This corresponds to the findings of previous studies. The previous research shows that the social organizations in China, especially the grassroots organizations in the early stages of their development, are usually small in scale and their operation largely relies on the strong will, organizational ability, personal contacts and social prestige of individual elites [33].

If building personal contacts and networking is the first step in establishing cooperation with the government, then a clear understanding of the position of social organizations in relation to the government is the key to strengthening the mutual trust. NW has identified over 30 organizations as key partners, who have several or even more than ten years of social service experience in the local society and have accumulated a local reputation through their practical work. Through the continuous cooperation with the government, they have won the trust of the relevant officials, and the local government departments have become their main source of funds. The social organizations have gradually defined their auxiliary role in relation to the government. Through assisting the government to achieve its social governance goals, the social organizations strengthened and improved their relationship with, and gained support from, the government. As the local judge and the head of social organization J said, “(Social organizations) shall not take themselves too seriously and claim to be a supplement to the government. Instead, we should say that we want to do something for the government”. Many of the heads of the social organizations pointed out that the social organizations should work discretely behind the government, overcome difficulties and help to solve problems for the government, even with no financial support. Only in this way can they be compensated with resources in return from the government, when the time comes.

In addition, once there is clarity about which government departments should become targets for mobilization, the social organizations need to adopt appropriate project design and embedding strategies in their efforts of administrative mobilization for the integration of sexuality education into specific government work. Evans [34] first put forward the concept of embedded autonomy to describe a specific relationship between the state and society. While the state maintains its autonomy, it can connect with the social groups through intermediary paths jointly constructed by institutionalized channels and social networks, so as to embed itself in society. Following that, many of the scholars used the embeddedness theory to analyze the interaction between the government and the social organizations. The previous research pointed out that the social organizations are mainly embedded in the government in terms of resources, legitimacy and institutional support, while the will and goals of the government are embedded in the operation of the social organizations to achieve the goal of improving governance performance [35,36].

The local social organizations investigated in this paper are experienced in undertaking projects through a government procurement process, and thereby have gained the support and trust of their local governments. Their mutually embedded relationship with the government worked smoothly. This paper described how the social organizations embed sexuality education into the regular work of the government. This research shows that, first, the social organizations can adopt the sexuality education program so that it is in line with the discourse system of their local governments and secondly, the project implementation can help the government accomplish specific tasks and achieve their performance objectives. Only in this way can the sexuality education programs gain support from the government.

The head of organization F has rich experience in packaging education programs to make them more acceptable to the government and the schools. He pointed out that CSE is not a familiar discourse for local governments, so it is essential for local social organizations, based on their understanding of the local government, to “translate” and transform it into a discourse that the local government departments can understand and accept, such as puberty education, self-protection education, mental health education, and so forth. The rebranding of sexuality education programs to meet the performance needs of different government functional departments has become a common implementation strategy for the local social organizations. Only when the re-branded sexuality education program can contribute to government performance, instead of increasing social risks and raising governmental concerns, will the local social organizations gain support and endorsement from the government in organizing the sexuality education training to meet the teaching targets of the program.

After three or four years of exploring program scaling-up, NW and its partner organizations have gradually formulated the basic idea of government mobilization to implement the sexuality education program in schools and communities. They believe that the local organizations should give priority to gaining support from the local civil affairs department and/or the education department. With more familiarity with the situation of communities and schools and greater power than other government functional departments on the matters relating to children and adolescents, these two departments are ideal partners to engage with, to promote sexuality education in schools regularly at institutional levels. Secondly, the social organizations can also rebrand and embed sexuality education into the regular work of other relevant government departments, such as the Health Department, Communist Youth League Committee, Women’s Federation, Science and Technology Bureau, the Commission for Industry and Commerce, the public security organs, procuratorial organs and People’s courts. This means that the local organizations should fully understand the needs of these departments. Another option is to integrate the sexuality education into the community-based social service projects, but NW thinks that this is not the ideal way to implement the program, because the number of children and adolescents participating in the community learning activities tends to be small. Besides, it is challenging to organize age-appropriate activities and deliver them continuously at the social project level, hence it is difficult to guarantee the teaching quantity and effectiveness.

All in all, with the empowering support provided by the public welfare ecosystem and with the accumulation of local resources and personal contacts, the local social organizations that identify with the large-scale implementation model are likely to remove the cultural and institutional obstacles to sexuality education through their mobilization strategy involving rebranding the education program to benefit the government’s performance and to achieve large-scale program implementation.

## 4. Discussion

This paper discusses how the social organizations, as an important social force to promote sexuality education for children and adolescents, deliver sexuality education across the country. Three important mechanisms emerged in the operation process of the sexuality education program: the standardized design of the sexuality education program; the organizational cooperation network and local administrative mobilization. Based on these three mechanisms, a top-down program chain has been formed (see Figure 1). This chain has expanded the resources that the social organizations can mobilize, enhanced social influence and could eventually break the various socio-cultural and institutional barriers to sexuality education to make way for large-scale program implementation.

At the same time, it should be noted that this program operation chain also increases the project management expenses for NW as the initiator of the sexuality education program. The long operation chain makes it difficult for NW to effectively control the program implementation and operation process of its partner organizations. A major program management issue is the compromised quality of the teaching. When NW first formulated the large-scale operation strategy, it hoped to improve the standard of program operation in four aspects: scale; effectiveness; quality consistency and per capita cost. Therefore, in terms of quality supervision, to assess the program implementation by its partner organizations, NW requires that the number of students in each class shall be kept between 20 to 30, that large-class teaching shall be kept to the minimum and systematic teaching be encouraged. The required class-size of 20–30 students is too strict, as a single class can easily have 40 to 50 students, and even up to 70 in certain areas. In the actual implementation, the frequency of the teaching and the class sizes were uneven. A common approach by the local partners was to deliver lectures in large combined classes and have a crowded period nearing the year-end to meet the teaching target, which affected the teaching quality. The extended school closure due to COVID-19 in the past few years made it even more challenging for the local social organizations to keep pace with meeting the teaching targets in schools.

In addition, this study notices that the local social organizations are mainly concerned about whether they meet the required teaching target in terms of student attendance, but for the time being, lack the awareness and ability to monitor and evaluate the quality of the teaching. Therefore, this paper recommends that NW, as the initiator and supporter of the program, should further consider how to verify and ensure the effectiveness of the program. While in the course of implementing the sexuality education program on a large scale, NW should also consider how to ensure the students’ sense of benefits gained, to sustain the teachers’ motivation and the ability to participate continuously and form a virtuous circle, which is an important embodiment of the value and vitality of the program and affects whether or not the children could acquire the necessary knowledge, attitudes and skills.

Finally, this paper discusses the promotion of sexuality education for children and adolescents in China beyond the perspective of program operationalization by social organizations. This paper puts forward three directions for researchers and sexuality education practitioners to further explore. First, sexuality education, as part of an overall quality education, plays an important role in promoting and ensuring students’ safety, physical and mental wellbeing and development. This investigation shows that the delivery of sexuality education enables students to feel more humanistic care, improves the teacher–student relationship, and enhances the students’ sense of belonging. Future research can verify these additional benefits through empirical analysis, which will help to build more social trust in sexuality education. Secondly, as far as the underage students are concerned, the lack of sexuality-related knowledge is the direct cause of some of the mental and physical harms related to sex. However, the sexuality-related problems faced by children and adolescents are socially constructed, and the solution depends on the improvement of the overall social environment, including relevant laws and regulations, as well as family support. Therefore, future research can further discuss how to promote cooperation among the different social subjects and build a systematic and effective protection mechanism for minors. Thirdly, the exploration of the social organizations in promoting sexuality education also provides inspiration for the government and the market to participate in the provision of the relevant services. For example, the social organizations play a major role in implementing the standardized sexuality education program. In comparison, the market is in a position to conduct research and develop creative sexuality education learning materials or software with its advantageous capital and manpower resources, to make sexuality education more attractive and easily implementable. Then, the government can take advantage of the knowledge products created by the social organizations and the market, and incorporate sexuality education into the school curriculum portfolio to benefit children and adolescents.

## Figures and Tables

**Figure 1 children-09-01474-f001:**
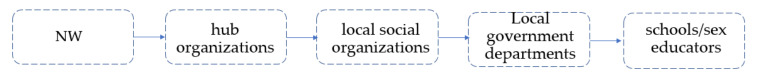
The top-down program chain.

**Table 1 children-09-01474-t001:** Information about social organizations (provincial, municipal, district-level and county-level are indicated to show the administrative divisions that the organizations are registered under).

Institution	Location	Type	Sources of Funds
NW	Xi’an, Shaanxi Province	Supporting platform	Foundation support
Hub organization A	Xi’an, Shaanxi Province	Provincial hub organization	Network crowdfunding
Hub organization B	Changsha, Hunan Province	Provincial hub organization	Government procurement
Hub organization C	Hefei, Anhui Province	Provincial hub organization	Foundation support
Hub organization D	Jinan, Shandong Province	Provincial hub organization	Government procurement
District-level organization E	Zaozhuang, Shandong Province	Implementing organization	Government procurement
County-level organization F	Linyi, Shandong Province	Implementing organization	Government procurement
municipal-level organization G	Weihai, Shandong Province	Implementing organization	Government procurement
County-level organization H	Weinan, Shaanxi Province	Implementing organization	Government procurement
District-level organization I	Xi’an, Shaanxi Province	Implementing organization	Government procurement
District-level organization J	Nanping, Fujian Province	Implementing organization	Government procurement
Municipal-level organization K	Hongjiang, Hunan Province	Implementing organization	Government procurement
District-level organization L	Siangtan, Hunan Province	Implementing organization	Government procurement
County-level organizations M	Zhuzhou, Hunan Province	Implementing organization	Government procurement
Municipal-level organization N	Jingmen, Hubei Province	Implementing organization	Government procurement
Municipal-level organization O	Zigong, Sichuan Province	Implementing organization	Government procurement
Municipal-level organization P	Wuhu, Anhui Province	Implementing organization	Government procurement

## Data Availability

Anonymized data are available on request from the corresponding author and with the permission of the participants in the study. The data presented in this study are not publicly available due to ethical reasons (to protect the privacy of participants).

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
