# Peer review of "Promoting Sexuality Education for Children and Adolescents on a Large Scale: Program Design, Organizational Cooperation Network and Administrative Mobilization"

_children, 2022, doi:10.3390/children9101474_

Round 1
Reviewer 1 Report
This is a really interesting study. Some changes will need to be made to make this publishable. Firstly, the paper needs some improvement in its English as sometimes word choices can be awkward. An example of this can be found on page four, where it is stated, "Based on the cognition of the concept of CSE put forward by the United Nations and referring to its content framework..." How does the word "cognition" fit in here? Do you mean "conceptualization"? As well, it seems as though CSE is being defined in relationship with the United Nations when in fact, there are many other places where CSE is defined and it was not the United Nations who founded the concept or approach.
I think more methodological information is needed. I find that at times it seems as though interviewers are being identified and I, as the reader, am left wondering if they gave permission to have themselves identified in the write-up of their data. As well, even when people are not named, information is provided about their title so that anyone who knows the field that these authors are writing on in a Chinese context might be able to identify the respondents.
I also find that the data write-up is lacking in that it mostly summarizes what participants said by paraphrasing and contains few direct quotes from the participants themselves. How were these data analyzed? I need more information about how these data were analyzed as the reader. It seems to skip from data collection to writing up the data.
I also noticed that the United Nations was mentioned a lot in the paper and then it was noted at the end that the United Nations funded the research. This seems like a conflict of interest.
I would like more information about how this research impacts actual children as I do not see this being discussed. Occasionally, teachers and educators are mentioned, but I'd like to see more implications for children in China discussed.
Reviewer 2 Report
The paper discusses the problems of promoting sexual education for the younger people in China and the role and mechanisms of social organizations. This is done through an empirical research aimed a identifying the main mechanisms beyond sexual educational programs and analyses the challenges posed by their operationalization.
The paper addresses a less known topic and it is therefore of the outmost interest. It is well structured and written, and it has an extended bibliography. Both the study methodology and the presentation of results are well managed.
A final remark is regarding the periods of the study. Especially in 2020, and especially in China, Covid has been an important variable in all aspect of our lives. I would have expected some comments or caveats regarding how it has impacted the sexual education programs, or an explanation of why it has not.
This having been said, I think that it could be published as it is, with just minor revisions regarding some typos, e.g. p. 2 line 13: there is a space between “ and Scaling; line 17: put a space between the indexed numbers 101112 -> 10 11 12; p. 5 line 13 from the bottom: do not put a come between subject and verb, ‘stipulates that, social..’ -> ‘stipulates that social’.
Reviewer 3 Report
This is a very well done study examining the implementation of sex education in China and the factors that contribute to successful integration of sex education into school programming. The multiple data sources including interviews with a variety of stakeholders, meeting notes, and direct observation led to illuminating key mechanisms that influence the provision of sex education at the local level.
I have two small suggestions for improving the paper. There are important observations about the importance of teacher comfort and skill with delivering sex education which appear only in the discussion section and not in the results section. I'd encourage the authors to move their data on teacher delivery of sex education into the results section as well.
The other suggestion is in the abstract and it is about word choice. Rather than using the word "targeted" I would suggest using "studied" or
"focused on".
I appreciate the rigorous work and well written study.
